# GSK2656157, a PERK Inhibitor, Alleviates Pyroptosis of Macrophages Induced by Mycobacterium *Bacillus Calmette–Guerin* Infection

**DOI:** 10.3390/ijms242216239

**Published:** 2023-11-12

**Authors:** Boli Ma, Xueyi Nie, Lei Liu, Mengyuan Li, Qi Chen, Yueyang Liu, Yuxin Hou, Yi Yang, Jinrui Xu

**Affiliations:** 1School of Life Sciences, Ningxia University, Yinchuan 750021, China; 15695013087@163.com (B.M.); 18794898774@163.com (X.N.); liuleidaisy@foxmail.com (L.L.); 20180044@nxmu.edu.cn (M.L.); chenq9510@163.com (Q.C.); 18395273708@163.com (Y.L.); 18209689779@163.com (Y.H.); 2Key Laboratory of Ministry of Education for Conservation and Utilization of Special Biological Resources in the Western, Ningxia University, Yinchuan 750021, China

**Keywords:** PERK, *Bacillus Calmette–Guerin*, macrophages, pyroptosis

## Abstract

Tuberculosis (TB) is the leading cause of human death worldwide due to *Mycobacterium tuberculosis* (Mtb) infection. Mtb infection can cause macrophage pyroptosis. PERK, as a signaling pathway protein on the endoplasmic reticulum, plays an important role in infectious diseases. It is not clear whether PERK is involved in the regulation of pyroptosis of macrophages during Mtb infection. In this study, *Bacillus Calmette–Guerin* (BCG) infection resulted in high expression of pro-caspase-1, caspase-1 p20, GSDMD-N, and p-PERK in the THP-1 macrophage, being downregulated with the pre-treatment of GSK2656157, a PERK inhibitor. In addition, GSK2656157 inhibited the secretion of IL-1β and IL-18, cell content release, and cell membrane rupture, as well as the decline in cell viability induced by BCG infection. Similarly, GSK2656157 treatment downregulated the expressions of pro-caspase-1, caspase-1 p20, caspase-11, IL-1β p17, IL-18 p22, GSDMD, GSDMD-N, and p-PERK, as well as reducing fibrous tissue hyperplasia, inflammatory infiltration, and the bacterial load in the lung tissue of C57BL/6J mice infected with BCG. In conclusion, the inhibition of PERK alleviated pyroptosis induced by BCG infection, which has an effect of resisting infection.

## 1. Introduction

Tuberculosis (TB) is a chronic respiratory infectious disease caused by *Mycobacterium tuberculosis* (Mtb) [1]. The co-infection of Mtb and human immunodeficiency virus (HIV) and the emergence of drug-resistant TB are important causes of human TB deaths [2,3]. Mtb infections can activate innate immunity, which plays an important antibacterial role in the early stages of Mtb infection [4]. Alveolar macrophages serve as first-line immune cells in the fight against Mtb infection.

Pyroptosis, a kind of programmed cell death driven on by proinflammatory mediators, is connected to the NOD-like receptor thermal-protein-domain-associated protein 3 (NLRP3) inflammasome [5,6]. The multienzyme complex known as the NLRP3 inflammasome is made up of NLRP3, the effector molecule pro-caspase-1, and apoptosis-related speckle-like protein (ASC) [7,8]. According to studies, when NLRP3 inflammasomes are activated, they assemble with the cysteine proteases caspase-1 and ASC, which causes caspase-1 to be cleaved and activated [9,10]. Gasdermin D (GSDMD) is cleaved by the inflammatory caspase-1 that NLRP3 activates, releasing GSDMD’s N-terminus. The plasma membrane is ruptured, and proinflammatory substances, such as interleukin (IL)-1β and IL-18, are released when holes are created in the plasma membrane by the N-terminus of GSDMD [11,12]. The morphology of pyroptosis shows cell swelling, plasma membrane rupture, and release of cytoplasmic contents [13]. Furthermore, the conversion of precursor 1β (IL-1β) and 18 (IL-18) into IL-1β and IL-18 is catalyzed by caspase-1 [14]. Mtb secretes a variety of virulence proteins to induce pyroptosis in host cells. For example, EST12 can bind with receptor for activated C kinase1 (RACK1) to activate NLRP3 inflammasome in macrophages to promote pyroptosis [15]. In addition, when Mtb was engulfed by phagosome, Mtb mediated pyroptosis of phagosome by triggering the cGAS/STING pathway via virulence factors [16]. To study the mechanism of macrophage pyroptosis during Mtb infection is helpful to provide a theoretical basis for the clinical treatment of Mtb.

Protein-kinase-like ER kinase (PERK) is a receptor protein on the membrane of the endoplasmic reticulum (ER) that monitors the aggregation of unfolded proteins in the ER [17]. The PERK-N segment is located within the ER membrane, and the C segment is located outside the membrane and contains kinase-binding domains. Normally, the inactive PERK-N segment binds to the ATPase domain of immunoglobulin heavy chain binding protein (Bip). When endoplasmic reticulum stress (ERS) occurs, the ATPase domain of Bip interacts with the luminal domain of PERK for dissociation, PERK dimerization, and activation [18]. PERK is involved in the regulation of inflammation, autophagy, and apoptosis in the course of pathogenic bacterial infection [19,20,21]. In recent years, PERK has been widely concerned due to its important role in the regulation of inflammation and cell death [22]. It has been confirmed that PERK is activated after macrophages are infected with Mtb [23]. Therefore, whether PERK plays a regulatory role in macrophage pyroptosis induced by Mtb infection remains unclear, which is important in revealing the pathogenesis of TB and controlling the progression of the disease. In this study, macrophages and C57BL/6J mice were treated with GSK2656157 and infected with BCG to reveal the regulatory role of PERK on pyroptosis, which would help to further understand Mtb infection mechanism and develop a more effective anti-Mtb strategy.

## 2. Results

### 2.1. BCG Infection Induced THP-1 Macrophage Pyroptosis

To investigate whether BCG infection causes pyroptosis, THP-1 macrophages were infected with BCG at MOI of 10 for 2, 6, 12, 24, and 48 h, and pyroptosis level was evaluated subsequently. The qRT-PCR results showed that the mRNA expression of *GSDMD* progressively increased with the time of BCG infection and reached the relatively higher level at 24 hpi (Figure 1A). The Western blot results showed that BCG infection significantly increased GSDMD-N protein expression, presented in an infection time-dependent manner, and reached the relatively higher level at 24 h post-infection (hpi) (Figure 1B,C). Additionally, the ELISA results showed that the secretion of IL-1β and IL-18 induced by BCG increased and presented an infection time-dependent manner, reaching the relatively higher secretion at 48 hpi (Figure 1D,E). The cell viability results detected by LDH kits and CCK-8 kits showed that BCG infection suppressed THP-1 macrophage viability in an infection time-dependent manner (Figure 1F,G). Taken together, these results indicated that BCG infection can induce pyroptosis of THP-1 macrophages.

### 2.2. BCG Infection Activates PERK of THP-1 Macrophage

To investigate the effect of BCG infection on the activation of the PERK, the BCG-infected macrophages were collected at 2, 6, 12, 24, and 48 h for analysis of PERK expression. The results of qRT-PCR showed that the mRNA expression of *PERK* was gradually increased in THP-1 macrophages at 2, 6, 12, 24, and 48 hpi (Figure 2A). Additionally, BCG induced a significant upregulation of p-PERK protein expression in THP-1 macrophages at 2, 6, 12, 24, and 48 hpi in a time-dependent manner (Figure 2B,C). These findings suggested that the PERK was activated in BCG-infected THP-1 macrophages.

### 2.3. GSK2656157 Suppressed BCG Infection-Induced Pyroptosis of THP-1 Macrophages

To explore the involvement of PERK in the BCG-induced pyroptosis of THP-1 macrophages, a PERK inhibitor (GSK2656157) was adopted to pretreat macrophages before BCG infection. The qRT-PCR results showed that GSK2656157 inhibited the mRNA expression of *GSDMD* and *pro-caspase-1* in BCG-infected THP-1 macrophages (Figure 3A,B). Western blot analysis showed that the blocking-up of PERK significantly inhibited the expression of GSDMD-N, pro-caspase-1, and caspase-1 p20 protein in BCG-infected THP-1 macrophages (Figure 3C–G). Furthermore, ELISA results showed that the inhibition of PERK by GSK2656157 significantly reduced the secretion of IL-1β and IL-18 induced by BCG (Figure 3H,I). Meanwhile, LDH and CCK-8 assay results showed that the GSK2656157 promoted the BCG-infected THP-1 macrophage survival (Figure 3J,K). In addition, the significant downregulation of GSDMD in BCG-infected THP-1 macrophages was detected by immunofluorescence because of the GSK2656157 inhibitory effect (Figure 3L,M). TEM results showed that GSK2656157 significantly alleviated BCG-infected THP-1 cell lysis, cell membrane rupture, and content release (Figure 3N). The above results suggested that pyroptosis of the THP-1 macrophages induced by BCG infection was regulated by PERK.

### 2.4. GSK2656157 Suppressed BCG-Infected Pyroptosis in the Lung of Mice

To investigate the effect of PERK upon BCG infection in vivo, mice were injected intraperitoneally with GSK2656157 following BCG challenge. The results of qRT-PCR showed that GSK2656157 inhibited the mRNA expression of the *GSDMD*, *pro-caspase-1*, and *caspase-11* in the lung tissue of BCG-infected mice (Figure 4A–C). GSK2656157 inhibited the protein expression of GSDMD-N, pro-caspase-1, caspase-1 p20, and caspase-11 in the lung tissue of BCG-infected mice (Figure 4D–I). Furthermore, IHC results directly showed that GSK2656157 inhibited p-PERK and GSDMD protein expression in the lung tissue of BCG-infected mice (Figure 4J–M). Collectively, these results demonstrated that PERK regulated pyroptosis in the lung tissue of BCG-infected mice.

### 2.5. GSK2656157 Inhibited the Expression of IL-1β and IL-18 in the Lung of BCG-Infected Mice

To further investigate whether PERK has a regulatory effect on inflammatory cytokine IL-1β and IL-18 in the lung tissue of BCG-infected mice, Western blot, qRT-PCR, and IHC assays were carried out for evaluation. The mRNA expressions of *pro-IL-1β* and *pro-IL-18* in the lung tissue were inhibited by GSK2656157 (Figure 5A,B). Similarly, Western blot results showed that GSK2656157 inhibited the protein expression of IL-1β p17 and IL-18 p22 in the lung tissue of BCG-infected mice (Figure 5C–E). Furthermore, IHC results showed that GSK2656157 inhibited the protein expression of IL-1β in the lung tissue of BCG-infected mice (Figure 5F,G). Combined with the above results in Section 2.4, these results further demonstrated that PERK is essential for pyroptosis in the lung tissue of BCG-infected mice.

### 2.6. GSK2656157 Attenuated the Level of Lung Damage, Inflammatory Infiltration, and Bacterial Colonization in BCG-Infected Mice

To further confirm the effect of the PERK on lung damage and the bacterial colonization in the lung of BCG-infected mice, HE and CFU assays were adopted, respectively. Mice infected with BCG showed some degree of fibrous tissue hyperplasia and inflammatory infiltrates. In contrast, the lungs of mice given BCG infection combined with GSK2656157 were less damaged and displayed less inflammatory infiltrates (Figure 6A,B). The bacterial load of BCG-infected mice treated with GSK2656157 in the lung was significantly reduced as compared with the BCG-infected mice (Figure 6C). These findings demonstrated that inhibition of PERK alleviated lung damage, inflammatory infiltration, and bacterial colonization in BCG-infected mice.

## 3. Discussion

The host’s immune system serves as a vital line of defense against Mtb. As the primary element of the innate immune system, alveolar macrophages are the initial effector cells to come into contact with Mtb and are important for disease progression and dissemination [24,25]. Mtb is the main pathogen of TB, and studying the interaction mechanism between Mtb and macrophages is of great significance for the treatment of TB. Mtb infection of macrophages has been reported to activate ERS and further regulate autophagy and apoptosis of macrophages [26,27]. However, whether ERS induced by Mtb infection mediates macrophage pyroptosis, and what the regulatory role of PERK is, remains unclear.

The nucleotide-binding leucine-rich-repeat-containing protein (or NOD-like receptors, NLRs) family consists of several members, and these proteins play different roles in innate immunity and inflammation [28]. A caspase recruitment domain (CARD) is a common structural motif of the NLRP protein family. It is responsible for the recruitment of ASC protein and pro-caspase-1 protein, as well as further activation of caspase-1 [29,30]. Caspase-1 mediates the proteolysis and activation of cytokines IL-1β and IL-18, and GSDMD. The resulting cell lysis of GSDMD is known as inflammation-induced pyroptosis. Pyroptosis plays an important role in eliciting proinflammatory responses to bring innate immune cells to the site of injury or infection. Pyroptosis is activated by various bacterial infections [31]. Mtb induces NLRP3 inflflammasome activation followed by pyroptosis, which causes severe damage and allows for the spread of Mtb [32]. NLRP3-deficient mice effectively control the Mtb infection [33]. In addition to NLRP3 mediating cell pyroptosis in Mtb infection, Mtb infection caused no effection in IL-1β secretion from NLRP12^-/-^ bone marrow-derived dendritic cells (BMDC) [28]. These results suggest that NLRP12 is not associated with pyroptosis induced by Mtb infection. It has been reported that the N-terminal of NLRP1b is cleaved by anthrax toxin to promote inflammasome activation and caspase-1-dependent pyroptosis [34,35]. This study provides important guidance for the mechanism study of NLRP1b in Mtb infection. Whether other NLRP family proteins are related to pyroptosis caused by Mtb infection needs further study. In this study, a model of Mtb-infected macrophages was established using BCG-infected THP-1 macrophages according to Chen et al. [36]. The absence of region of difference 1 (RD1) is the major reason that BCG is different from Mtb, but it is still valuable to use BCG to infect macrophages to reveal the mechanism of Mtb infection [37,38]. Our results demonstrated that BCG infection upregulated the expression of GSDMD-N, as well as the secretion of IL-1β and IL-18, and suppressed THP-1 macrophage viability in an infection time-dependent manner. These results indicate that BCG infection induces THP-1 macrophage pyroptosis.

Under stress conditions, the ER initiates the unfolded protein response (UPR) to restore homeostasis. PERK is one of the three transmembrane sensors of UPR and culminates in the transcriptional regulation of gene expression [39]. Most studies have focused on the role of PERK-mediated autophagy and apoptosis in neurodegenerative diseases, metabolic diseases, and cancer [18,40], but there are few studies on PERK mediated pyroptosis in infectious diseases. Our results demonstrated that BCG infection activates PERK as well as pyroptosis. PERK inhibitor GSK2656157 inhibited the expression of pyroptosis markers GSDMD-N, pro-caspase-1, and caspase-1 p20; reduced the secretion of IL-1β and IL-18; alleviated cell content release and cell membrane rupture; and attenuated the cell death of THP-1 macrophages induced by BCG infection. These results suggest that PERK has a regulatory role in the pyroptosis of BCG-infected THP-1 macrophages.

To further explore the role of PERK in the lung of BCG infection, we employed BCG-infected C57BL/6J mice treated with GSK2656157. The results showed that GSK2656157 inhibited expressions of GSDMD-N, pro-caspase-1, caspase-1 p20, caspase-11, IL-1β, and IL-18 in the lung tissue of BCG-infected mice. Meanwhile, inhibition of PERK alleviated lung damage and inflammatory infiltration and reduced the bacterial load in the lung tissue of BCG-infected mice. Other studies have shown similar results, with PERK having an inhibitory effect on pyroptosis. GSK2656157 reduced the LDH release of 66cl4 cells treated with metabolite trimethylamine N-oxide (TMAO) [41]. A recent study has also confirmed that the inhibition of PERK can reduce the expression of caspase-1, IL-1β, and IL-18 in SH-SY5Y cells [42]. ERS exacerbates acute pancreatitis by promoting caspase-1-dependent pyroptosis via the PERK pathway [43].

Some previous studies proved that downstream substrates of PERK, C/EBP homologous protein (CHOP), and nuclear respiratory factor 2 (Nrf2) are involved in the regulation of pyroptosis [44,45]. CHOP and Nrf2 are regulated by eukaryotic translation initiation factor 2 alpha (eIF2-α), which is phosphorylated by phosphorylated PERK (p-PERK) [39]. Therefore, whether PERK mediating CHOP and/or Nrf2 plays a similar function during Mtb infection and its mechanism needs further investigation.

## 4. Materials and Methods

### 4.1. BCG, Cell Lines, Antibodies, Reagents, and Animals

BCG (Chengdu Institute of Biological Products, Chengdu, China); THP-1 cells (Chinese Academy of Sciences Cell Bank, Beijing, China; RPMI-1640 medium (C11875500BT, Gibco, Carlsbad, CA, USA); fetal bovine serum (FBS) (10099141C, Gibco, Carlsbad, CA, USA); phorbol 12-myristate 13-acetate (PMA) (P1585, Sigma-Aldrich, St. Louis, MO, USA); Protease Inhibitor Cocktail (78442, Sigma-Aldrich, St. Louis, MO, USA); the reverse transcription kit (Vazyme Biotech, Nanjing, China); *PerfectStart*^TM^ Green qPCR SuperMix kit (AQ601-01-V2, Beijing TransGen Biotech, Beijing, China); Trizol (15596018CN, ambion, Austin, TX, USA). GSK2656157 (HY-13820, MCE, Monmouth Junction, NJ, USA); BCA protein quantitative detection kit (KGP902, KGI, Nanjing, China); protein extraction reagent M-PER™ (78505, Thermo Fisher Scientific, Waltham, MA, USA); β-mercaptoethanol (M8211, Solarbio, Beijing, China); anti-β-actin (20536-1-AP, Proteintech, Hamilton, IL, USA); anti-Pro-caspase-1 (22915-1-AP, Proteintech, Wuhan, China); anti-mouse GSDMD (66387-1-Ig, Proteintech, Wuhan, China); HRP-goat anti-rabbit/mouse IgG (SA00001-2/SA00001-1, Proteintech, Wuhan, China); FITC-goat anti-rabbit IgG (SA00003-2, Proteintech, Wuhan, China); anti-rabbit GSDMD (39754S, CST, Danvers, MA, USA); anti-phospho-PERK (Thr982) antibody (DF7576, Affinity, Austin, TX, USA); Anti-IL-1β (ab254360, abcam, Cambridge, UK); anti-caspase-11 (ab180673, abcam, Cambridge, UK); hematoxylin dye solution (Servicebio Biotech, Wuhan, China). Eosin dye (bomeibio, Hefei, China). Immunohistochemical SP kit (ZSGB Biological Technology, Beijing, China). Lactate dehydrogenase (LDH) test kit (ab102526, Abcam, Cambridge, UK). Cell counting kit-8 (CCK-8) test kits (abs50003, Absin, Shanghai, China); the ELISA kits for the detection of IL-1β and IL-18 (EK0392/EK0864, Boster Biological Technology, Wuhan, China). Female C57BL/6J mice (Beijing Charles River Experimental Technology, Beijing, China).

### 4.2. THP-1 Cell Culture and BCG Infection

THP-1 cells were maintained in RPMI-1640 medium contained with 10% FBS in a 5% CO_2_ incubator at 37 °C. THP-1 cells were seeded into a six-well plate at 1.0 × 10^6^ cells/well and stimulated with 50 ng/mL PMA for 48 h to differentiate into macrophages [46]. BCG culture was performed based on a previously described method [47]. After reaching 80–90% confluence, the THP-1 macrophages were infected with BCG at 10 multiplicities of infection (MOI) for 2, 6, 12, 24, and 48 h. In the co-treatment experiment, THP-1 macrophages were pretreated with PERK inhibitor GSK2656157 (5 μM) for 2 h before BCG infection.

### 4.3. Western Blotting Analysis

The mice lungs or cultured cells infected with BCG were lysed on ice by protein extraction reagent M-PER™ containing protease inhibitors. The total protein concentration was determined using the BCA protein quantitative detection kit. Samples were separated by 10% SDS-PAGE and then transferred to polyvinylidene difluoride (PVDF) membranes. The PVDF membrane was blocked with 5% skimmed milk in Tris-buffered saline with Tween-20 (TBST) at room temperature for 2 h and incubated with the corresponding primary antibody overnight at 4 °C. HRP-goat anti-rabbit/mouse IgG antibody was incubated at room temperature for 2 h before being exposed to ECL chromogenic solution (Amersham Imager600 Automatic Chemiluminescence Imager, General Electric, Boston, MA, USA). Image-Pro Plus 6.0 software was used to evaluate and quantify grayscale data, which were then normalized to β-actin levels. At least three replications of each experiment were conducted.

### 4.4. Quantitative Real-Time PCR (qRT-PCR)

The mice lungs or cultured cells infected with BCG were extracted with Trizol reagent, and cDNA was synthesized by reverse transcription according to the reverse transcription kit’s instructions. For qRT-PCR analysis, the *PerfectStart*^TM^ Green qPCR SuperMix detection kit was used to amplify cDNA by the Quantity Studio 5 real-time fluorescence quantitative PCR instrument (Thermo Fisher Scientific, Waltham, MA, USA). For a total of 40 cycles, the reaction procedure was circulated for 5 s at 95 °C and 30 s at 60 °C. 2^−ΔΔCt^ was used to calculate the relative mRNA expression levels of each gene. Table 1 displays the pertinent information about the primers used.

### 4.5. ELISA

ELISA was performed to evaluate the concentration of IL-1β and IL-18 in culture supernatants of BCG-infected macrophages. The assay was performed using human IL-1β and IL-18 ELISA kits (Boster, Wuhan, China), following the manufacturer’s instructions. An Enspire Fluorescent Enzyme Label Instrument (PerkinElmer, MA, USA) was used to measure the absorbance at OD_450nm_, and the results were calculated.

### 4.6. LDH

The cell culture supernatants were collected to analyze extracellular LDH release according to the manufacturer’s instructions. The absorbance was measured at 450 nm using a microplate reader.

### 4.7. CCK-8

The cells grown to the logarithmic phase were plated into a 96-well plate with 1.0 × 10^4^ mL^−1^ cells/well. After BCG infection for 24 h, 10 μL CCK-8 was added to each well and incubated for an additional 1–4 h. The absorbance was measured at 450 nm using a microplate reader, and the data were recorded. Cell viability (%) = (experimental group OD value − blank group OD value)/control group OD value × 100%.

### 4.8. Immunofluorescence Analysis

THP-1 cells were seeded into 12-well plate and stimulated with PMA for 48 h in advance, pretreated with the PERK inhibitor GSK2656157 for 2 h, and inoculated with BCG at MOI of 10 for 24 h. The THP-1 macrophages were subsequently fixed with 4% paraformaldehyde (PFA) for 30 min and permeabilized with 0.5% Triton-X 100 for 30 min. After blocking with 5% donkey serum for 1 h, the GSDMD primary antibody was incubated overnight. Finally, FITC-goat anti-rabbit IgG antibody and DAPI (for nuclear counterstaining) were added. Cells were observed and imaged using a fluorescent confocal microscope (Thermo Fisher Scientific, Waltham, MA, USA).

### 4.9. Transmission Electron Microscopy (TEM)

THP-1 cells were fixed with 3% glutaraldehyde, then with 1% osmium tetroxide, and then progressively dehydrated with acetone. Epoxy resin 812 was used for embedding. After slicing, uranium acetate and lead citrate were stained. Images were collected using JEM-1400-FLASH transmission electron microscopy.

### 4.10. Animal Challenge Experiments

All female mice at 6 weeks of age were randomly divided into the control group (n = 6), the BCG infection group (n = 6), and the BCG + GSK2656157 treatment group (BCG + GSK, n = 6). Experiments were started after the arrival of the mice in our animal facility with food and water ad libitum and regular 12:12 light–dark cycle. BCG was diluted to 1 × 10^7^ colony-forming units (CFU)/50 μL per animal suspension using PBS [48]. Three different treatment protocols were administered: (1) Mice from the BCG-infected group were intranasally instilled into the respiratory tract of the ether-anesthetized mice. Control mice received the same volume of vehicle (PBS). (2) In addition to BCG respiratory instillation, mice from BCG combined with the GSK2656157 treatment group were intraperitoneally injected with GSK2656157 (50 mg/kg/d) for 28 days [49]. Control mice were injected with same volume of vehicle (PBS). (3) Mice in the control group were cultured normally. The mouse lungs were collected individually for Western blot, qRT-PCR, HE, immuno-histochemistry (IHC), and CFU assay. All animal experiments were approved by the Technology Ethics Committee of Ningxia University and were in accordance with the guidelines of the Animal Welfare Council of China.

### 4.11. H&E Staining

H&E staining was performed according to the instructions provided by the reagent manufacturer (Solarbio, Beijing, China). The lung tissue of mice was fixed, embedded in paraffin, and then sliced. The sections were dewaxed with xylene for 5 min before being soaked in anhydrous ethanol (95, 80, and 70%) for 2 min each. Sections were stained for 10 min with hematoxylin solution and soaked in differentiation solution for 30 s. After 5 min in 50 °C water, the sections were stained with eosin solution for 1 min. The sections were dehydrated in a graded alcohol solution before being cleared with xylene. The pathological conditions were scored after the images were collected using an Olympus IX73 microscope (Olympus, Tokyo, Japan).

### 4.12. IHC

Lung samples from BCG-infected, BCG + GSK2656157-treated, and control mice were collected, fixed in 4% PFA for over 24 h at 4 °C, then embedded in paraffin. Sections were blocked using the subsequent procedure. Briefly, sections were soaked in 3% H_2_O_2_ for 30 min to quench endogenous peroxidase and then blocked with goat serum. Sections were incubated with anti-p-PERK, GSDMD, and IL-1β antibody overnight at 4 °C. Sections were washed again then incubated with HRP-goat anti-rabbit/mouse IgG antibody. After washing, sections were incubated with diaminobenzidine (ZSGB, Beijing, China) for color development then photographed and analyzed using an Olympus IX73 microscope (Olympus, Tokyo, Japan).

### 4.13. Bacterial Colony-Forming Unit (CFU) Count

The lung tissue fluid prepared by mechanical grinding was spread on series diluted medium on 7H10 agar plates and incubated at 37 °C for 3 weeks. CFU values were calculated based on the number of colonies on the plates. The dilution factor and the plate count were multiplied to work out the final bacterial CFU value.

### 4.14. Statistical Analysis

All experimental data were collected three times independently and analyzed with GraphPad Prism software. One-way ANOVA was used to compare differences between multiple groups, and the *t*-test was used to compare differences between two groups. * *p* < 0.05, ** *p* < 0.01, *** *p* < 0.001 were considered statistically significant.

## 5. Conclusions

In summary, we found that BCG infection activated PERK, induced pyroptosis, and inhibition of PERK-attenuated pyroptosis. GSK2656157 downregulated the pyroptosis related protein expressions in the lung tissue of BCG-infected C57BL/6J mice and decreased damage and inflammatory infiltration in the lung of BCG-infected mice.

## Figures and Tables

**Figure 1 ijms-24-16239-f001:**
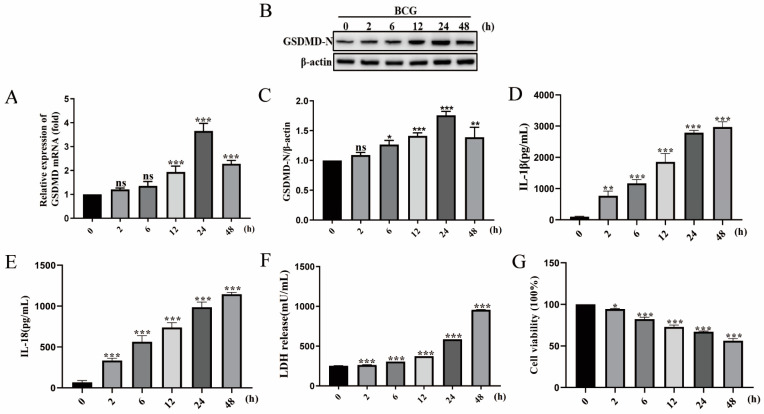
BCG infection induced THP-1 macrophage pyroptosis. The THP-1 macrophages were infected with BCG at MOI of 10 for 0, 2, 6, 12, 24, and 48 h. (**A**) qRT-PCR results of *GSDMD* mRNA expression. (**B**) Western blot results of GSDMD-N protein expression. (**C**) GSDMD-N protein expression analysis histogram. (**D**,**E**) ELISA results of IL-1β (**D**) and IL-18 (**E**) secretion levels. (**F**) LDH kits detection results. (**G**) CCK-8 assay results of cell viability. The gray value of a Western blot was determined using ImageJ. ^ns^
*p* > 0.05, * *p* < 0.05, ** *p* < 0.01, *** *p* < 0.001 (one-way ANOVA analysis).

**Figure 2 ijms-24-16239-f002:**
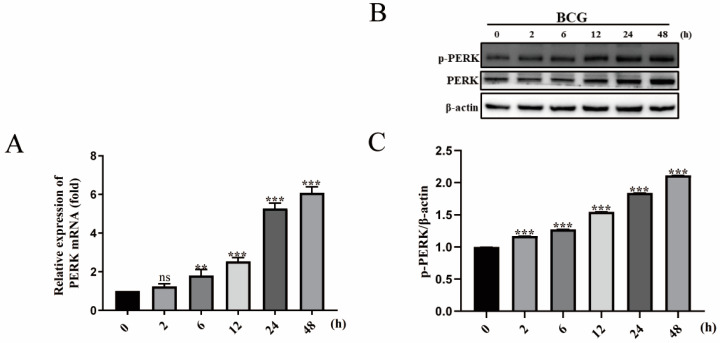
BCG infection activated PERK of the THP-1 macrophage. The THP-1 macrophages were infected with BCG at MOI of 10 for 0, 2, 6, 12, 24, and 48 h. (**A**) qRT-PCR results of *PERK* mRNA expression. (**B**) Western blot assay results of p-PERK and PERK protein expression. (**C**) p-PERK protein expression analysis histogram. ^ns^
*p* > 0.05, ** *p* < 0.01, *** *p* < 0.001 (one-way ANOVA analysis).

**Figure 3 ijms-24-16239-f003:**
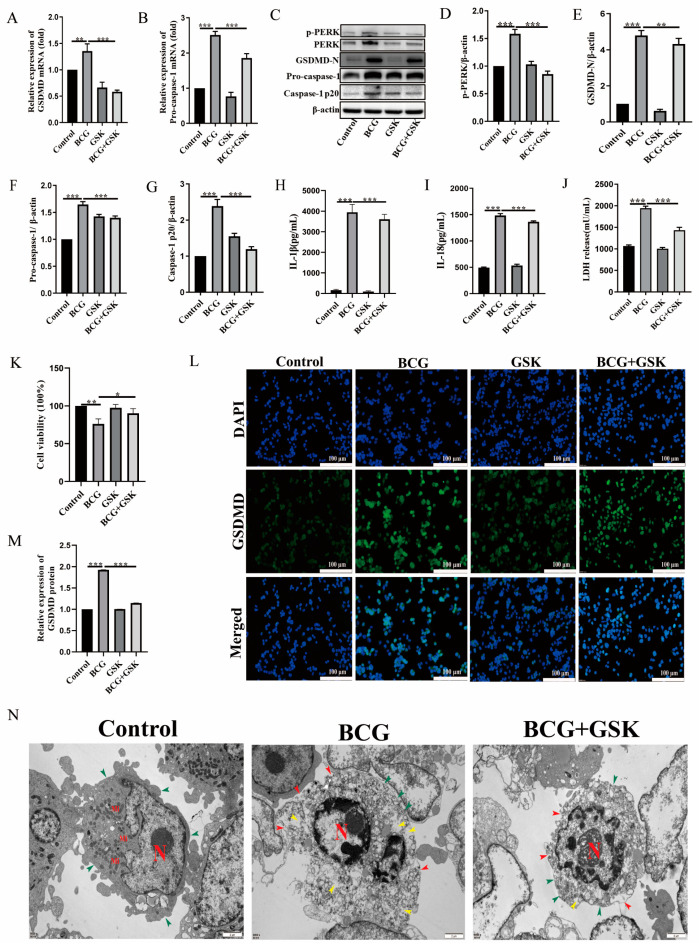
GSK2656157 suppressed BCG-infection-induced pyroptosis of THP-1 macrophages. The THP-1 macrophages were pretreated with GSK2656157 for 2 h and then infected with BCG at MOI of 10 for 24 hpi. (**A**,**B**) qRT-PCR results of *GSDMD* (**A**) and *pro-caspase-1* (**B**) mRNA expression in control, BCG, GSK, and BCG + GSK groups, respectively. (**C**) Western blot analysis of p-PERK, PERK, GSDMD-N, pro-caspase-1, and caspase-1 p20 protein expression levels in control, BCG, GSK, and BCG + GSK groups, respectively. (**D**–**G**) Expression histogram of p-PERK (**D**), GSDMD-N (**E**), pro-caspase-1 (**F**), and caspase-1 p20 (**G**) protein. (**H**,**I**) ELISA results of IL-1β (**H**) and IL-18 (**I**) secretion levels in control, BCG, GSK, and BCG + GSK groups, respectively. (**J**) LDH kit detection results in control, BCG, GSK, and BCG + GSK groups. (**K**) CCK-8 assay results of cell viability in control, BCG, GSK, and BCG + GSK groups. (**L**) Immunofluorescence staining results of GSDMD protein expression in control, BCG, and BCG + GSK groups; GSDMD is green, DAPI is blue, and the scale was 20 μm. (**M**) GSDMD protein analysis histogram. (**N**) The results of cell morphology photographed by TEM in control, BCG, and BCG + GSK groups. Scale bar: 2 μm. Nucleus (**N**), mitochondria (Mi), intact and continuous cell membranes (green arrow), broken cell membranes (red arrow), cell contents released (yellow arrow). * *p* < 0.05, ** *p* < 0.01, *** *p* < 0.001 (one-way ANOVA analysis).

**Figure 4 ijms-24-16239-f004:**
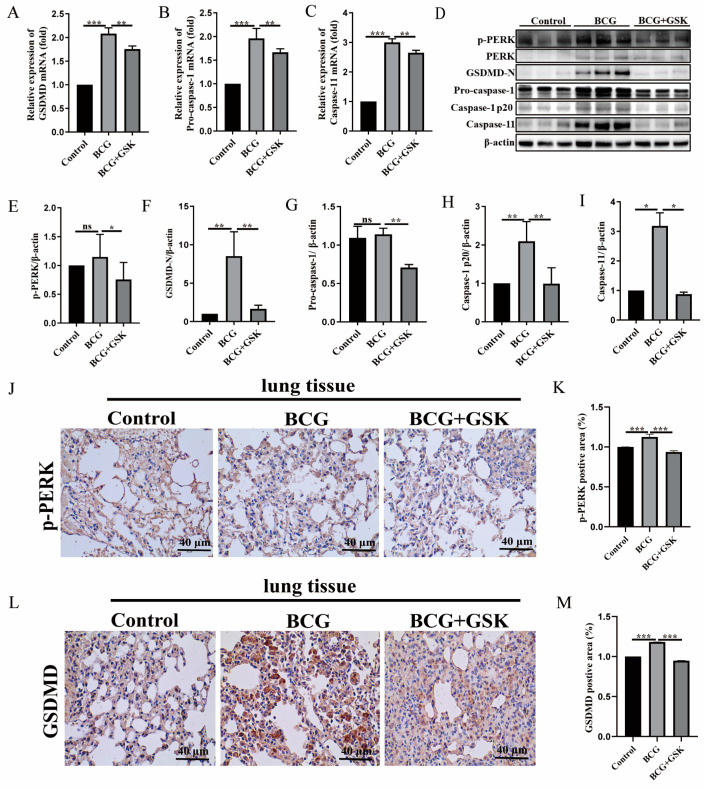
GSK2656157 inhibited pyroptosis in the lung tissue of BCG-infected C57BL/6J mice. The C57BL/6J mice were infected with BCG and then treated with GSK2656157. (**A**–**C**) qRT-PCR results *GSDMD* (**A**), *pro-caspase-1* (**B**), and *caspase-11* (**C**) mRNA expression in control, BCG, and BCG + GSK groups, respectively. (**D**) Western blot analysis of p-PERK, PERK, GSDMD-N, pro-caspase-1, caspase-1 p20, and caspase-11 protein expression in control, BCG, and BCG + GSK groups, respectively. (**E**–**I**) Expression histogram of p-PERK (**E**) GSDMD-N (**F**), pro-caspase-1 (**G**), caspase-1 p20 (**H**), and caspase-11 (**I**) protein. (**J**–**M**) IHC staining results of lung tissues with p-PERK (**J**) and GSDMD (**L**) protein expression in control, BCG, and BCG + GSK groups, respectively; the scale is 40 μm, ^ns^
*p* > 0.05, * *p* < 0.05, ** *p* < 0.01, *** *p* < 0.001 (one-way ANOVA analysis).

**Figure 5 ijms-24-16239-f005:**
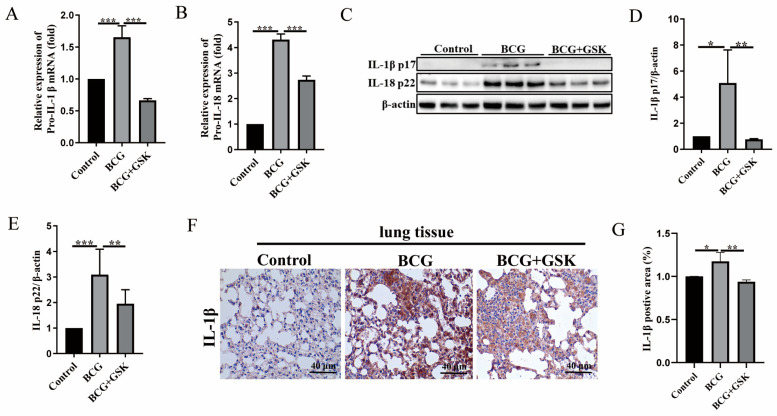
GSK2656157 reduced the expression of IL-1β p17 and IL-18 p22 in the lung tissue of BCG-infected C57BL/6J mice. The C57BL/6J mice were infected with BCG and then treated with GSK2656157. (**A**,**B**). qRT-PCR results of *pro-IL-1β* (**A**) and *pro-IL-18* (**B**) mRNA expression in control, BCG, and BCG + GSK groups, respectively; (**C**) Western blot analysis of IL-1β p17 and IL-18 p22 in control, BCG, and BCG + GSK groups, respectively. (**D**,**E**) IL-1β p17 (**D**) and IL-18 p22 (**E**) protein expression analysis histogram. (**F**,**G**) IHC results of IL-1β protein expression in control, BCG, and BCG + GSK groups; the scale is 40 μm. * *p* < 0.05, ** *p* < 0.01, *** *p* < 0.001 (one-way ANOVA analysis).

**Figure 6 ijms-24-16239-f006:**
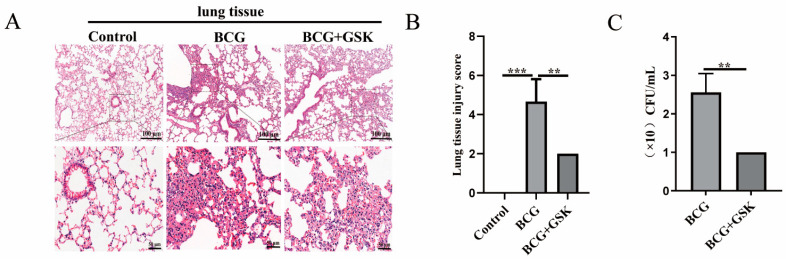
GSK2656157 inhibited fibrous tissue hyperplasia, inflammatory infiltration, and bacterial load in the lung tissue of BCG-infected C57BL/6J mice. The C57BL/6J mice were infected with BCG and then treated with GSK2656157. (**A**) HE staining results of fibrous tissue hyperplasia and inflammatory infiltration in control, BCG, and BCG + GSK groups. (**B**) Statistical results of lung tissue injury scores in C57BL/6J mice; the scale is 40 μm. (**C**) Analysis results of bacterial load in the lung tissue of C57BL/6J mice; ** *p* < 0.01, *** *p* < 0.001 (one-way ANOVA analysis).

**Table 1 ijms-24-16239-t001:** Primer sequences in qRT-PCR.

Gene	Forward Primer	Reverse Primer
Human-β-actin	CCTGGCACCCAGCACAAT	GGGCCGGACTCGTCATAC
Human-GSDMD	AGCCCTACTGCCTGGTGGTTAG	CACGCTGCACGTCTGGTTCC
Mouse-β-actin	GTGCTATGTTGCTCTAGACTTCG	ATGCCACAGGATTCCATACC
Mouse-pro-caspase-1	ATACAACCACTCGTACACGTCTTGC	TCCTCCAGCAGCAACTTCATTTCTC
Mouse-GSDMD	CGATGGGAACATTCAGGGCAGAG	ACACATTCATGGAGGCACTGGAAC
Mouse-IL-1β	CTGTCGGACCCATATGAGCTGAAAG	TGTCGTTGCTTGGTTCTCCTTGTAC
Mouse-IL-18	GGCTGCCATGTCAGAAGACTCTTG	AGTGAAGTCGGCCAAAGTTGTCTG
Mouse-caspase-11	GACTTAGGCTACGATGTGGTGGTG	ATGTGCTGTCTGATGTCTGGTGTTC

## Data Availability

All the data are included in the main manuscript.

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
