# Peer review of "GSK2656157, a PERK Inhibitor, Alleviates Pyroptosis of Macrophages Induced by Mycobacterium Bacillus Calmette–Guerin Infection"

_ijms, 2023, doi:10.3390/ijms242216239_

Round 1
Reviewer 1 Report
Comments and Suggestions for Authors
Intracellular bacteria pathogens, including M. tuberculosis the causative agent of tuberculosis, or for example, L. pneumophila, are serious threats to health. Those bacteria once internalized by macrophages hijack several macrophages' signaling pathways, including the possibility that the phagosome fuses with lysosome/s, thus enabling the bacteria to grow and proliferate intracellularly. In the recent past, we have witnessed a steady increase in antibiotic resistance worldwide and this represents a serious warning alarming the scientific community, including clinicians and surgeons. In their manuscript titled "GSK2656157, a PERK inhibitor alleviates pyroptosis of macrophages induced by mycobacterium Bacillus Calmette-Guerin infection" the authors the authors assessed the effects of a PERK inhibitor, GSK2656157, both in vitro and in vivo, on BCG infection. Basically, from their inquiry, they conclude that BCG infection activates PERK and induces pyroptosis. Inhibition of PERK weakens pyroptosis and brings beneficial effects to the lung tissue by reducing inflammatory responses. Though the manuscript presents some intriguing hints before publication some issues should be revised.
- When the authors show Western blots they are kindly invited to provide, alongside p-PERK, also the appropriate control represented by PERK itself. Consistently, quantification should be done on PERK and not actin.
- When it comes to Fig. 2 I am a little bit skeptical. When comparing the IF and the quantification the pictures do not seem to be representative of the quantification. Indeed, the difference highlighted by the quantification can not be appreciated from the picture. I would also advice the authors to include picture of the GSK by itself condition, as control.
- Throughout the whole manuscript, I would suggest modifying the Figures by swapping the order of the panels and placing mRNA panels and then the protein ones. Indeed, it is quite likely that if mRNA levels are low the protein ones might follow the same pattern. Is it not?
- The section "Material and Methods" requires a few amendments (e.g. line 432: how long macrophages are incubated with PMA to differentiate; it seems that Western blot bands quantification details are missing; the authors should clarify and detail clearer the animal treatment protocol -lines 488-490-. What do they mean when they say "...continuously intraperitoneally injected..."? All day long?).
- A few typos are scattered throughout the main text and should be edited (e.g., lines 412, 431, etc...).
Comments on the Quality of English Language
The English language is quite fine, just a few things should be amended (e.g. line 479 cells are permeabilized not permeated; line 516, what do the authors mean with "...lung tissue grounded..."?; and other few are scattered throughout the main text).
Reviewer 2 Report
Comments and Suggestions for Authors
1. Does Mycobacterium like other bacteria activate other NLRPs such as NLRP1b and can activate pyroptosis which happens in other bacteria. Please discuss these and cite relevant literature (PMID: 28137237, 30872533).
2. In Figure 2 and Figure 3 the pPERK blot imagesa re different in Figure 2 it is one band and Figure 3 it is two bands. Literature ssearch it is usually two bands. So please repeat the experiment in Figure 2A and provide new figure.
3. In some IHC images the control image chosen has less cells. Please provide a figure with similar cell density.
4. In Figure 3L microscopy data, better image is needed to see the GSDMD expression.
Round 2
Reviewer 1 Report
Comments and Suggestions for Authors
In the revised version of the manuscript titled "GSK2656157, a PERK inhibitor alleviates pyroptosis of macro-phages induced by mycobacterium Bacillus Calmette-Guerin infection" the authors addressed all the issues raised by the reviewer. I really thank the authors for the efforts made. Consistenly, I consider the current version of the manuscript suitable for publication in IJMS.